# Polymer Thin Film Promotes Tumor Spheroid Formation via JAK2-STAT3 Signaling Primed by Fibronectin-Integrin α5 and Sustained by LMO2-LDB1 Complex

**DOI:** 10.3390/biomedicines10112684

**Published:** 2022-10-24

**Authors:** Sunyoung Seo, Nayoung Hong, Junhyuk Song, Dohyeon Kim, Yoonjung Choi, Daeyoup Lee, Sangyong Jon, Hyunggee Kim

**Affiliations:** 1Department of Biotechnology, College of Life Science and Biotechnology, Korea University, Seoul 02841, Korea; 2Department of Biological Sciences, Korea Advanced Institute of Science and Technology (KAIST), Daejeon 34141, Korea

**Keywords:** cancer stem cells, polymer X, STAT3, reporter system, fibronectin, integrin, LMO2

## Abstract

Cancer stem-like cells (CSCs) are considered promising targets for anti-cancer therapy owing to their role in tumor progression. Extensive research is, therefore, being carried out on CSCs to identify potential targets for anti-cancer therapy. However, this requires the availability of patient-derived CSCs ex vivo, which remains restricted due to the low availability and diversity of CSCs. To address this limitation, a functional polymer thin-film (PTF) platform was invented to induce the transformation of cancer cells into tumorigenic spheroids. In this study, we demonstrated the functionality of a new PTF, polymer X, using a streamlined production process. Polymer X induced the formation of tumor spheroids with properties of CSCs, as revealed through the upregulated expression of CSC-related genes. Signal transducer and activator of transcription 3 (STAT3) phosphorylation in the cancer cells cultured on polymer X was upregulated by the fibronectin-integrin α5-Janus kinase 2 (JAK2) axis and maintained by the cytosolic LMO2/LBD1 complex. In addition, STAT3 signaling was critical in spheroid formation on polymer X. Our PTF platform allows the efficient generation of tumor spheroids from cancer cells, thereby overcoming the existing limitations of cancer research.

## 1. Introduction

With recent advances in cancer biology, significant improvements have been made in clinical diagnosis and cancer treatment. However, successful anti-cancer treatments remain restricted due to cancer recurrence, metastasis, and cell resistance towards therapeutics [1,2,3]. Recent studies have suggested that cancer stem-like cells (CSCs) are a subpopulation of cancer cells responsible for posing challenges to the complete surgical resection of tumors [4,5]. CSCs share several molecular and biological characteristics with normal stem cells [4,5]. However, multiple signaling pathways mediating the survival, proliferation, and differentiation of normal stem cells are aberrantly regulated in CSCs [4,6]. Notch, Wnt, Hedgehog, nuclear factor-κB (NF-κB), Janus kinase/signal transducers and activators of transcription (JAK/STAT), and tumor growth factor-β (TGF-β) signaling pathways play crucial roles in maintaining the population and characteristics of CSCs [4,6].

To identify potential targets for cancer therapy, the isolation and characterization of CSCs in primary tumors of patients are crucial. Thus, methods to isolate CSCs from primary tumors using cell surface markers or specific cell culture conditions have recently been developed [7,8,9]. For example, cellular surface markers such as CD24, CD44, CD133, epithelial cell adhesion molecule (EpCAM), leucine-rich-repeat-containing G-protein-coupled receptor 5 (LGR5), and CD117 have been used to isolate CSCs from cancer cells using fluorescence-activated cell sorting (FACS) and magnetic cell sorting [10,11,12,13]. In addition, serum-free culture conditions supplemented with various combinations of growth factors have been applied to enrich or maintain isolated CSCs from patients diagnosed with cancer [14,15,16].

Despite the current efforts, the use of patient-derived CSCs involves several limitations. First, the majority of human cancers harbor rare populations of CSCs (<1–2%) [17,18,19,20]. Second, CSC markers identified to date have not been fully established. Third, only a few CSCs express the identified CSC markers [21].

A functional polymer thin-film (PTF) platform was, therefore, developed to overcome these limitations [22]. Poly (2,4,6,8-tetravinyl-2,4,6,8-tetramethyl cyclotetrasiloxane), designated as pV4D4, induced the transformation of cancer cells to tumorigenic and drug-resistant CSCs [22]. Non-CSCs can dedifferentiate and acquire stem cell-like properties owing to their plasticity [23,24]. Several cellular signaling pathways, such as Wnt, TGF-β, and JAK-STAT, can drive cancer cell dedifferentiation and enhance stem cell-like properties [25,26,27].

In this study, we evaluated the functionality of a modified version of a previous PTF platform, pV4D4, which had issues with mass production. This version was designated as polymer X. In addition, we identified the cell-signaling mechanisms underlying tumor spheroid formation under polymer X culture conditions.

## 2. Materials and Methods

### 2.1. Cell Lines and Culture Condition

The human ovarian cancer cell line SKOV3 was purchased from the Korean Cell Line Bank. The cells were cultured and maintained in the Roswell Park Memorial Institute-1640 (RPMI-1640) medium (Gibco, Grand Island, NY, USA) supplemented with 10% fetal bovine serum (FBS) (HyClone, Logan, UT, USA), 1% penicillin/streptomycin (HyClone), and 25 mM HEPES (Gibco).

Human glioblastoma (GBM) cell lines A1207, LN18, LN229, T98G, and U87MG and the human cervical cancer cell line HeLa were purchased from the American Type Culture Collection (ATCC). These cell lines were cultured and maintained in Dulbecco’s modified Eagle’s medium (DMEM) with high glucose (4500 mg/L) (HyClone), supplemented with 10% FBS, 1% penicillin/streptomycin, 2 mM L-glutamine, and 50 µg/mL gentamicin (Tocris, Bristol, UK). All cells were incubated and maintained at 37 °C in a humidified 5% CO_2_ chamber.

### 2.2. Spheroid Formation on pV4D4 and Polymer X and Treatment of Inhibitors

Cells were seeded at a density of 2 × 10^6^ cells (100 mm dish) or 0.6 × 10^6^ cells (60 mm dish) on pV4D4-coated plates or polymer X-coated plates. RPMI-1640 or DMEM, as appropriate, containing 10% Knockout Serum Replacement (SR) (Gibco) 1% penicillin/streptomycin and 25 mM HEPES, was used as culture medium. For optimum spheroid growth, the culture medium was replaced every 2 days. Spheroids were dissociated with an accutase solution (Sigma-Aldrich, St. Louis, MO, USA) every 4 days to prevent cell death at the spheroid core.

The focal adhesion kinase (FAK) inhibitor 14, Y15 (7.5 μM) (Cat. #3414, Tocris) was used to inhibit integrin-mediated activation of FAK. JAK inhibitor I (1 μM) (Cat. #420099, Calbiochem, Ct, San Diego, CA, USA) and AG-490 (200 μM) (Cat. #0414, Tocris) was used to prevent the phosphorylation of JAK. Nifuroxazide (50–100 μM) (cat. #481984, Calbiochem) and S3I-201 (50 μM) (Cat. #S1155, Selleckchem, Houston, TX, USA) were used to prevent phosphorylation of STAT3.

### 2.3. Plasmids

For the reporter assay, transcription factor response elements (22 × TAT3 response element: 5′-TGCTTCCCGAACGT-3′, 5 × CSL response element: 5′-CCGTGGGAA-3′, 4 × Smad binding element: 5′-CAGACA-3′, 4 × TCF/LEF1 response element: 5′-ATCAAAGG-3′, 2 × NF-κB response element: 5′-GGGACTTTCCGCTGGGGACTTTCC-3′, 8 × GLI response element: 5′-GAACACCCA-3′) were cloned into the pLL-mCMV-EGFP-PGK-Blast vector.

### 2.4. Lentiviral Transduction and Gene Transfections

For lentivirus preparation, each reporter gene vector was transfected with second-generation lentiviral packaging plasmids, ∆8.9 and VSV-G, using the LipoJet transfection reagent (SignaGen Laboratories, Rockville, MD, USA) into an immortalized human kidney cell line, HEK293FT cells (ATCC). The pLL-mCMV-EGFP-PGK-Blast vector was used as a control. The transformed lentiviruses were harvested after 48 h post-transfection, incubated with the Lenti-X concentrator (Clontech Laboratories, Mountain View, CA, USA), and centrifuged. The cells were then infected with the transformed lentiviruses for 24 h in the presence of 6 μg/mL hexadimethrine bromide (Sigma-Aldrich) for 24 h.

Small-interfering RNAs (siRNAs) targeting *ITGA5*, *LDB1*, and *LMO2* were purchased from Sigma Aldrich. siRNAs targeting *FN1*, *JAK1*, and *JAK2* were designed using the siDirect version 2.0 software [28]. The ScreenFect™ A transfection reagent (Wako Pure Chemical, Osaka, Japan) was used for cellular transfection, according to the manufacturer’s instructions. The siRNA sequences are listed in Appendix A.

### 2.5. Establishment of Reporter Gene-Introduced Cell Lines

SKOV3 cells were infected with lentiviruses containing each reporter gene vector and selected using blasticidin (Cayman Chemical, Ann Arbor, MI, USA). The cells were harvested and washed twice with 1× phosphate buffer saline (PBS). Cells were re-suspended in 1× PBS containing 1% FBS and analyzed using a flow cytometry system (BD FACSAria). Green fluorescent protein (GFP)-negative cells were sorted and used for subsequent experiments.

### 2.6. Immunocytochemistry/Immunofluorescence (ICC/IF)

Eight-day-cultured SKOV3 spheroids of 100~200 μm were fixed with 4% paraformaldehyde (PFA) (Sigma-Aldrich) for 30 min at room temperature and incubated in 15% sucrose followed by 30% sucrose in PBS overnight at 4 °C. Thereafter, spheroid samples were frozen embedded in the OCT compound (Leica, Wetzlar, Germany) in cryomolds and stored at −80 °C until use. Sample blocks were sliced into 10-μm-thick sections via the Cryostat (Leica) instrument. After rinsing with distilled water and blocking with 3% BSA/1× PBS, they were stained with rabbit anti-human fibronectin primary antibody (ab2413; Abcam) overnight at 4 °C, followed by Alexa Fluor^®^ 647 donkey anti-rabbit secondary antibody (ab150075; Abcam) for 1 h at room temperature. The sections were counterstained with Hoechst 33342 for 10 min and mounted with Mounting Medium (Dako, Santa Clara, CA, USA) for confocal (Carl Zeiss, Oberkochen, Germany) or fluorescent microscopy (Nikon, Minato City, Tokyo, Japan). Captured images were further processed and analyzed using ImageJ (https://imagej.nih.gov accessed on 8 June 2021), and average fluorescence (FL) intensity data were calculated for 5 spheroids and normalized to that of the ultra-low attachment (ULA) (Corning, Corning, NY, USA) spheroid control group.

### 2.7. RNA Extraction and Quantitative RT-PCR

Total RNA was extracted from cells using the QIAzol lysis reagent (Qiagen, Hilden, Germany) according to the manufacturer’s instructions. RNase-free and DNase-treated RNA (1 µg) were used as a template to synthesize cDNA using the RevertAid First-Strand cDNA Synthesis Kit (Thermo Fisher Scientific, Waltham, MA, USA). Quantitative Reverse Transcription-PCR (qRT-PCR) analysis was performed on an iCycler IQ real-time detection system (Bio-Rad, Hercules, CA, USA) using IQ Supermix with SYBR Green (TaKaRa Bio, Shiga, Japan). Gene expression was quantified using the standard^2−∆∆Ct^ method as described previously [22,29]. The expression levels of the target genes were normalized to those of 18S rRNA. Primers used for qRT-PCR amplification are listed in Appendix A.

### 2.8. Western Blot Analysis

For Western blots, whole-cell extracts were prepared using the radioimmunoprecipitation assay (RIPA) lysis buffer (LPS solution, Daejeon, Korea) comprised of 150 mM NaCl, 1% NP-40, 0.1% SDS, and 50 mM Tris (pH 7.4) containing 1 mM NaF, 1 mM Na_3_VO_4_, 1 mM ß-glycerophosphate, 2.5 mM sodium pyrophosphate, and protease inhibitor cocktail (Roche, Basel, Switzerland). Proteins were quantified using the Bradford assay reagent (Bio-Rad) according to the manufacturer’s instructions. Proteins (20–30 μg) were separated on 8–12% sodium dodecyl sulfate polyacrylamide gel electrophoresis (SDS-PAGE) and immunoblotting was performed as described above. The primary antibodies used were as follows: Anti-pSTAT3 (Y705) (1:500, 9145S; Cell Signaling Technology, Danvers, MA, USA), anti-STAT3 (1:1000, 4904S; Cell Signaling Technology), anti-fibronectin (1:500, ab6328; Abcam, Cambridge, UK), anti-integrin α5 (1:1000, ab150361; Abcam), anti-JAK1 (1:500, 50996S; Cell Signaling Technology), anti-JAK2 (1:1000, 3230S; Cell Signaling Technology), anti-GP130 (1:2000, ab226346; Abcam), anti-LDB1 (1:1000, ab96799; Abcam), anti-LMO2 (1:500, NB110-78626SS; Novus Biologicals, Littleton, CO, USA), and anti-α-tubulin (1:10,000, T6199; Sigma Aldrich). α-Tubulin was used as a loading control.

### 2.9. Reporter Gene Assay

GFP-negative cells were used for the reporter gene assay. Cells were seeded at a density of 6.0 × 10^5^ cells (6-well plate) or 2.0 × 10^6^ cells (100 mm plate) and allowed to form spheroids using the method described earlier. After 8 days of culture, the intensity of the GFP signal was measured using IncuCyte Zoom (Sartorius, Version; 2016 B).

### 2.10. In Silico Analysis

To investigate the differentially regulated cellular pathways between cells cultured on the tissue culture plate (TCP) and polymer X-coated plate, gene set enrichment analysis (GSEA) was performed using the MsigDB Hallmark gene signature (https://www.broadinstitute.org/gsea accessed on 15 February 2020) [30]. ClueGO, a Cytoscape plugin for the visualization of functionally organized gene ontology/pathway networks, was used to determine the enriched biological terms in spheroids cultured on polymer X-coated plates [31]. The GO analysis was repeated on a published expression dataset (GSE213872).

### 2.11. Quantification and Statistical Analysis

All experiments were performed at a minimum of three independent replicates. Data were analyzed using a two-tailed Student’s *t*-test and presented as mean ± standard error of the mean (SEM). The level of statistical significance stated in the text was based on the *p*-values. a: *p* < 0.05; b: *p* < 0.01; and c: *p* < 0.001 were considered statistically significant.

## 3. Results

### 3.1. Polymer X Allowed Formation of Uniform Spheroids and Activated Cellular STAT3 Signaling

To verify the utility of polymer X for the enrichment of tumor spheroids (Figure 1A), we investigated the functionality of polymer X employing an established cancer cell line (Figure 1B). The human ovarian cancer cell line SKOV3 was seeded on a tissue culture plate (TCP) or a polymer X-coated plate. After 8 days of culture, SKOV3 cells formed tumor spheroids on polymer X without additional supplementary factors, but not on TCP (Figure 1B).

CSCs can self-renew and form spheres in serum-free culture. Sphere-forming ability is one of the indicators of the stem cell-like properties of cancer cells [32,33,34]. Thus, we verified the expression of phenotypic markers of CSCs on spheroids cultured on TCP and polymer X-coated plates (Figure 1C). The cellular mRNA expression of *LGR5*, *CD44*, sex-determining region Y-box 2 (*SOX2*), *NANOG*, Aldehyde dehydrogenase 1 (*ALDH1A1*), *CD117*, and *EpCAM* was upregulated in polymer X-cultured spheroids relative to TCP-cultured cells. Thus, cells culturing on polymer X induced the formation of tumor spheroids, which expressed CSC-associated genes.

To investigate the signaling pathways that contributed to CSC-associated marker upregulation in polymer X-induced spheroids, we constructed a reporter gene assay system (Figure 1D). Reporter system vectors contain promoter elements responsible for Notch, Wnt, Hedgehog, NF-κB, STAT3, and TGF-β signaling pathways. The activation of each signaling pathway was identified using GFP expression. Reporter system-introduced SKOV3 cells were cultured on polymer X for 8 days (Figure 1E). Interestingly, the GFP signal was observed in the tumor spheroids introduced with the STAT3 reporter system. In addition, GSEA showed that STAT3 signaling was more enriched in spheroids cultured on polymer X as compared to those cultured on TCP (Figure 1F).

STAT3 has two important phosphorylation sites, Tyr705 phosphorylated by JAK, and Ser727 phosphorylated by serine/threonine kinase [35]. Phosphorylation of Tyr705 is critical for the nuclear translocation of STAT3, therefore regulating STAT3 signal activation. Therefore, we performed Western blotting to investigate the phosphorylation status of STAT3 in cells cultured on polymer X including previously used pV4D4 (Figure 1G, Appendix A. During spheroid culturing on polymer X, the cells were harvested every 24 h for a total culturing period of 8 days to observe STAT3 phosphorylation (pY705-STAT3; p-STAT3) over time. An upregulation of p-STAT3 was observed at 24 h of culture. Additionally, mRNA expression of target genes of the STAT3 signaling pathway, interleukin-6 (*IL-6*), early growth response protein 1 (*EGR1*), p21-activated kinase 2 (*PAK2*), and prostaglandin-endoperoxide synthase 1 (*PTGS1*) was analyzed (Figure 1H). Upregulation in the cellular expression of *EGR1* and *PTGS1* was observed, which was in agreement with the activation of the protein STAT3. Collectively, polymer X induced tumor spheroid formation ability and STAT3 signaling activation in the candidate cancer cells.

### 3.2. Initial Activation of STAT3 Signaling Was Induced by Fibronectin-JAK2 Axis

IL-6 is a well-known canonical STAT3 signaling inducer [35]. Interestingly, while STAT3 activation was observed for cells cultured on polymer X, the mRNA expression of *IL-6* in these cells decreased (Figure 1H). Therefore, to identify alternative mechanisms inducing STAT3 activation in these cells, gene ontology (GO) analysis using the Cytoscape software add-on ClueGO plugin was conducted on differentially expressed genes (DEGs) that were upregulated in polymer X-cultured spheroids and the results were compared to those for the TCP-cultured cells (Figure 2A). The extracellular matrix (ECM)-related ontology was enriched in the tumor spheroids obtained using polymer X. This observation was in agreement with the findings of a previous study (Figure 2A) [22]. Compared to the tumor spheroids cultured on an ultra-low attachment (ULA) plate, a large amount of ECM was enriched inside those cultured on polymer X (Figure 2B). Additionally, immunofluorescence (IF) staining revealed high expression of fibronectin 1 (FN1) on the tumor spheroids cultured on polymer X (Figure 2B).

Fibronectin has been reported to mediate STAT3 activation [36,37]. Therefore, we cultured cells on polymer X and investigated the levels of cellular proteins associated with the fibronectin and STAT3-signaling pathway (Figure 2C). The protein level of FN1 in these cells increased from day 1 of THE culture and this increase was maintained for 8 days. siRNA-mediated *FN1* knockdown during the culture period caused a reduction in the levels of STAT3-related proteins on days 4 and 8 (Figure 2D). In addition, based on the fluorescence intensity measured using the IncuCyte system, *FN1* knockdown compromised the spheroid forming ability of the cells cultured on polymer X. Accordingly, the intensity of GFP indicated that the STAT3 reporter activity also reduced (Figure 2E).

Fibronectin binds to integrin and activates focal adhesion kinase (FAK) and SRC [38,39]. As a result, FAK and SRC mediate the outside-in signal transduction [40]. To verify whether the phosphorylation of STAT3 was mediated by the activation of FAK-SRC-mediated signaling, FAK inhibitor 14 was added during the polymer X culture condition, and the levels of STAT3-related proteins were analyzed (Appendix A). The results showed that while the level of p-SRC decreased, that of p-STAT3 remained unchanged. We therefore hypothesized that canonical integrin-FAK signaling was not a mediator of STAT3 activation in cells cultured on polymer X [36,37].

Additionally, although the levels of STAT3-related proteins revealed maximum reduction on day 8 of the culture, the JAK2 level decreased significantly on day 4 (Figure 2C). JAK transmits an upstream ligand-receptor signal to STAT3 by phosphorylating the Tyr705 residue of STAT3. Several previous studies showed that STAT3 can be activated by integrin via JAK2 [36,41]. To determine whether phosphorylation of STAT3 can be mediated via activation of the fibronectin-JAK2 axis, we knocked down *JAK2* using siRNA and analyzed the levels of STAT3-related proteins. The levels of STAT3-related proteins decreased on day 4 of the culture. This observation was in agreement with that recorded for *FN1* knockdown in cells (Figure 2D). Additionally, the results displayed that the levels of STAT3-related proteins and tumor spheroid formation decreased when cultured on polymer X supplemented with JAK inhibitor I (Appendix A). In contrast to JAK2, JAK1 is not affected by fibronectin but is related to the STAT3 signaling pathway. To evaluate the effect of JAK1 on the activation of STAT3-related proteins, we knocked down *JAK1* using siRNA (Appendix A). As a result, the level of STAT3-related proteins decreased to some extent but did not show any significant changes compared to that observed as a result of the *JAK2* knockdown. Therefore, we concluded that early phosphorylation of STAT3 could be mediated by fibronectin-activated JAK2.

In addition, as fibronectin generally regulates cells through integrin receptors, we examined whether integrin signaling was involved in the activation of JAK2 via fibronectin [42,43]. Since integrin α5β1 is a major receptor of fibronectin, we knocked down integrin subunit alpha 5 (*ITGA5*) in the SKOV3 cells using siRNA and identified the levels of STAT3-related proteins. The levels of all STAT3-related proteins, including JAK2, decreased when compared to cells transfected with non-target siRNA (Figure 2G). Therefore, we concluded that the tumor spheroids cultured on polymer X were activated by the fibronectin-integrin-JAK2-STAT3 signaling axis.

### 3.3. Long-Term Activation of STAT3 Signaling via LMO2-LDB1 Complex

In our study, polymer X-cultured cells displayed fibronectin expression that increased from day 1 with the phosphorylation of STAT3, whereas the expression levels of JAK2 were increased from day 1, but gradually decreased over time.

Therefore, we hypothesized that there may be other factors that render p-STAT3 continuously active. According to our previous study, the phosphorylation of STAT3 can be induced by the cytoplasmic LMO2-LDB1 complex [44]. For cells cultured on polymer X, the levels of LMO2 and LDB1 gradually increased from the time of culture until day 8 (Figure 3A).

Unlike the observations recorded post-*FN1* knockdown, the knockdown of *LMO2* did not cause a reduction in the levels of STAT3-related proteins on day 4 of the culture. This observation was accompanied by an insignificant change in the LMO2 level on day 4 of culture. In contrast, the level of LMO2 significantly increased on day 8. Hence, it was not surprising to observe a significant decrease in the levels of STAT3-related proteins on day 8 as a result of *LMO2* knockdown (Figure 3B). Similar results were observed on *LDB1* knockdown (Figure 3C). In addition, *LMO2* or *LDB1* knockdown caused significant reductions in GFP intensity and the spheroid-forming ability of the cells (Figure 3D). Taken together, these results indicated that STAT3 activation and tumor spheroid formation were sustained by LMO2 and LDB1.

### 3.4. Polymer X-Induced Tumor Spheroids Acquired Cancer Stem-Like Properties via STAT3 Signaling

To examine the significance of upstream genes involved in STAT3 activation regulating properties of tumor spheroids, we knocked down *FN1*, *ITGA5*, *JAK1*, *JAK2*, *LDB1*, and *LMO2* from the cells. As a result, the mRNA expression of the CSC-associated genes was significantly decreased (Figure 4A). Since early STAT3 activation was regulated by fibronectin-JAK2, and its long-term activation was maintained by the cytoplasmic LMO2-LDB1 complex, phosphorylation of STAT3 on the cells cultured on polymer X might play an important role in regulating cancer stem-like properties in these cells.

We also performed direct STAT3 inhibition to examine its effects on the CSC properties of the candidate cells. The results showed that tumor spheroid formation and the expression of CSC-related genes decreased, even when cultured on polymer X supplemented with inhibitors. We confirmed that various p-STAT3 inhibitors decreased the levels of p-STAT3 in the cells cultured on polymer X (Figure 4B,C), therefore reducing the spheroid-forming ability of these cells (Figure 4D). In addition, the mRNA expression of both STAT3 target genes and CSC-related genes notably decreased (Figure 4E,F).

Moreover, when tumor spheroids cultured on polymer X were transferred to TCP, the levels of STAT3-related proteins, in particular, LMO2 and LDB1, decreased on day 2 of the transfer (Figure 4G). Subsequently, the mRNA expression of CSC-associated markers also markedly decreased (Figure 4H). Collectively, these results indicated that the STAT3 signaling pathway played an important role in the activation of cancer stemness in cancer cells grown on polymer X.

### 3.5. General Application of Polymer X Using Various Cancer Cell Lines

To determine the application of polymer X to other cancer cell lines, human GBM cell lines A1207, LN18, U87MG, LN229, and T98G and the human cervical cancer cell line HeLa were cultured on polymer X and tested for their tumor spheroid formation ability. Similar to the SKOV3 cells, polymer X-cultured GBM cells and HeLa cells formed better spheroids as compared to those cultured on TCP (Figure 5A). In addition, we attempted to determine whether the formation of tumor spheroids was mediated by the activated STAT3 signaling pathway in the cells cultured on polymer X. In line with SKOV3 ovarian cancer cells, STAT3 signaling was also activated in tumor spheroids formed by the various GBM cells and HeLa cells cultured on polymer X (Figure 5B–D). Therefore, the formation of tumor spheroids and STAT3 signaling activation may be a general phenomenon in polymer X-based cell cultures.

Taken together, culturing established cancer cell lines on polymer X would be of great help in the study of cancer biology. In addition, it allows the use of tumor spheroids that exhibit cellular physiological conditions similar to those present in tumors in vivo.

## 4. Discussion

In this study, we elucidated that when cultured on the newly revised PTF platform, polymer X, the STAT3 signaling pathway was activated to obtain CSC-like properties in tumor cells. In particular, the STAT3 signaling pathway, which was activated through the fibronectin-JAK2 axis, was continuously maintained by the cytoplasmic LMO2-LDB1 complex when cultured on polymer X.

CSCs have been identified as one of the predominant causes of inducing therapeutic resistance in cancer cells. The importance of cells with cancer stem-like properties has been, therefore, constantly emerging among researchers [5,45,46,47]. Despite the significance of CSCs, the low availability of patient-derived CSC populations makes it difficult to apply them to research on cancer biology and the discovery of therapeutic targets [5,45]. In addition, the commonly used TCP enriched with various supplements is not sufficient to maintain the original characteristics of CSCs derived from patients [48,49]. To overcome these limitations, in this study, we proposed PTF-coated plates that can be utilized as a useful platform to obtain cancer stem-like populations. These plates provide an efficient and convenient platform to re-acquire cancer stem-like characteristics from differentiated cancer cells without any additional modulation.

Although our study failed to extensively investigate the mechanism of STAT3 activation via the fibronectin-JAK2 signaling pathway during the early time point, we considered that one plausible reason for STAT3 activation might be due to cellular stress during low cell attachment to the surface of polymer X. Another possible cause might be mechanotransduction, which includes the concept that enriched ECM molecules can be converted to biochemical signals that provoke adaptive transcriptional and other cellular responses [50,51]. Fibronectin is one of the most common ECM components and is a well-known regulator of mechanotransduction signaling due to the conformational flexibility of its complex [52,53,54]. Cancer cells are more likely to sense and respond to their physical contact with ECM and the neighboring cells, which improves the cell cycle, epithelial-hepatic metastasis, and cellular motility [52,55]. Changes in the enrichment of ECM proteins modulate the physical environment and forces exerted on the cell and are transmitted to the nucleus, eliciting biochemical and transcriptional responses in cells [50,51,56]. Interestingly, in our study, fibronectin was highly enriched in tumor spheroids obtained by culturing cells on polymer X (Figure 2B). It is noteworthy that when cultured on polymer X, the cells did not attach very well to the bottom surface, similar to a ULA plate. In our previous study, tumor spheroid formation was compared between pV4D4 and ULA [22]. Both plates did not allow efficient cell attachment to the bottom of the plate. However, compared to ULA, PTF-coated plates provided enriched ECM, therefore allowing cancer cells to form uniformly sized tumor spheroids. These results were consistent with the results of the RNA sequencing analysis reported in one of our previous studies [22].

We also demonstrated the role of the activated STAT3 signaling pathway in mediating tumor spheroid formation by cells cultured on polymer X. Many studies have shown that differentiated cancer cells can acquire cancer stem-like features by de-differentiation via various genetic alterations and disruption of their epigenetic status [25,26,57,58]. Thus, the phenotypic changes in the tumor spheroids obtained by culturing differentiated cancer cells, including established cancer cell lines on polymer X, may explain the serial de-differentiation processes. Research on the biological and molecular mechanisms of the cellular acquisition of stem-like properties is essential to understanding the phenotypic changes induced by polymer X. While adequate information on additional phenotypic changes in tumor spheroids obtained from the actual polymer X is also required, our results provide preliminary mechanistic insights into phenotypic changes in cancer cells induced by polymer X.

According to our previous study, the cytoplasmic LMO2-LDB1 complex enhanced STAT3 activity to maintain CSC characteristics in cancer cells [44]. However, culturing cells on polymer X cannot clearly explain how the protein level of LMO2 gradually increased with time, or how the STAT3 signaling pathway was maintained by the LMO2-LDB1 complex. In our study, the knockdown of *JAK2* and *ITGA5*, including fibronectin (upstream factors involved in the initial tumor spheroid formation) mediated significant reductions in the levels of LMO2 and LDB1 (Appendix A). This finding might be attributed to a feedback loop system, which can be described in two ways: A biological phenomenon in which the output either amplifies (positive feedback) or inhibits (negative feedback) the system [59,60]. Accordingly, we assumed that after p-STAT3 signaling was activated, the cytoplasmic LMO2-LDB1 complex was formed as a result of the increased level of LMO2 through positive feedback to maintain the cancer stem-like properties of the cancer cells. Taken together, we revealed that tumor spheroids obtained using our newly designed polymer X culture platform acquired cancer stem-like characteristics as revealed through the activated STAT3 signaling pathway, which was initially mediated via the fibronectin-JAK2 axis and the subsequent maintenance of the increased level of p-STAT3 by the LMO2-LDB1 complex.

In conclusion, polymer X can be used as a platform for identifying and targeting various characteristics of CSCs and establishing potential multi-directional anti-cancer therapeutic approaches. Therefore, further studies on the application of polymer X for culturing not only established cancer cell lines but also patient-derived cancer cells would shed light on advances in cancer biology.

## Figures and Tables

**Figure 1 biomedicines-10-02684-f001:**
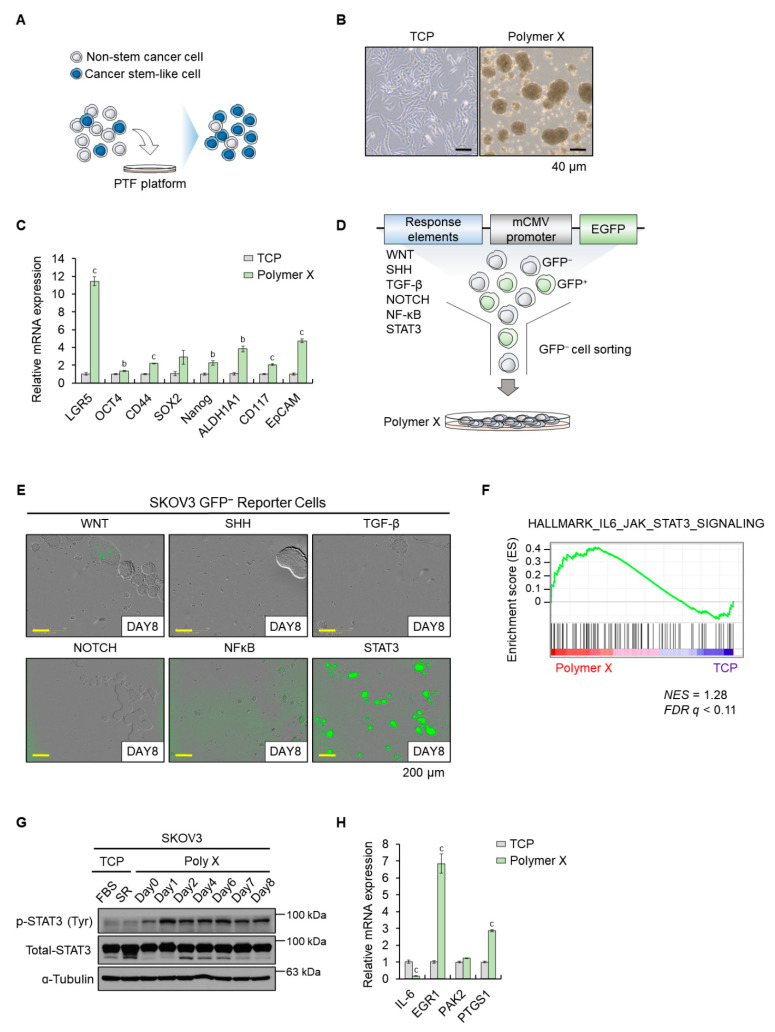
Acquisition of sphere formation ability and stemness signaling in the SKOV3 cells grown under polymer X culture condition. (**A**) Schematic illustration indicating the formation of tumor spheroid on polymer X. (**B**) Morphologies of SKOV3 cancer cells when cultured on TCP or polymer X for 8 days (d). Scale bar, 40 μm. (**C**) The mRNA levels of each indicated gene compared between cells cultured on TCP versus polymer X, as determined by real-time PCR. Data are expressed as mean ± SEM. Two-tailed Student’s *t*-test was used to analyze the statistical significance between each group (*n* = 3 for each group). b: *p* < 0.01; c: *p* < 0.001. (**D**) A schematic diagram showing GFP-negative population in cells cultured on polymer X and sorted by establishing a reporter system. (**E**) Image obtained by culturing each GFP-negative cell line transduced with each reporter system vector as indicated on polymer X for 8 days using IncuCyte system. Scale bar, 200 μm. (**F**) Gene set enrichment analysis (GSEA) displayed JAK-STAT3 signaling signature enrichment in SKOV3 cancer cells cultured on polymer X. (**G**) Cell lysates from SKOV3 cancer cells cultured on TCP or polymer X by date were immunoblotted with antibodies specific to pY705-STAT3 (p-STAT3), total STAT3, and α-tubulin. (**H**) The mRNA levels of the STAT3 target genes compared between cells cultured on TCP versus polymer X, as determined by real-time PCR. Data are expressed as mean ± SEM. Two-tailed Student’s *t*-test was used to analyze the statistical significance between each group (*n* = 3 for each group). c: *p* < 0.001.

**Figure 2 biomedicines-10-02684-f002:**
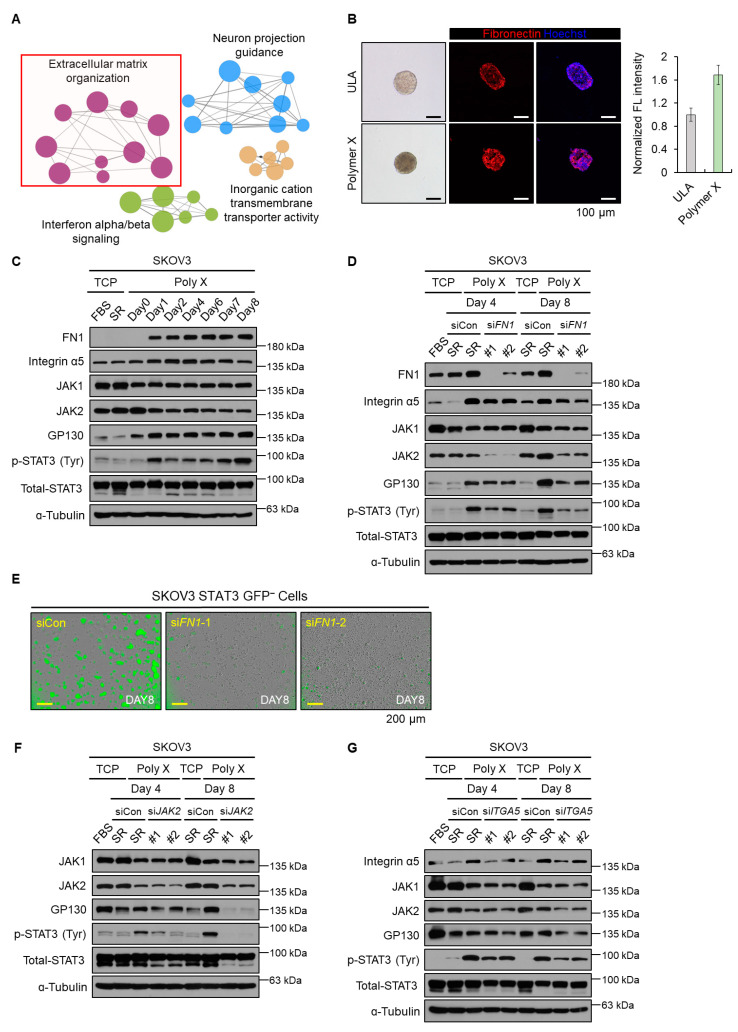
Initial activation of STAT3 signaling was induced by fibronectin-JAK2 axis. (**A**) Functional network analysis of Gene Ontology (GO). Significantly enriched GO terms were visualized using the Cytoscape software add-on ClueGO plugin. The Benjamini–Hochberg false discovery rate was used to analyze statistical significance. Each node represents a significantly enriched GO term. (**B**) Image obtained by SKOV3 cancer cells cultured on polymer X or ULA for 8 days, and representative image displaying IF representing fibronectin expression in 8 days of SKOV3-derived tumor spheroids cultured on polymer X or ULA. Scale bar, 100 μm. (**C**) Cell lysates from SKOV3 cancer cells cultured on polymer X by date were immunoblotted with antibodies specific to FN1, integrin ɑ5, JAK1, JAK2, GP130, p-STAT3, total STAT3, and α-tubulin. (**D**) Cell lysates from SKOV3 cancer cells transfected with either non-target siRNA or si*FN1* cultured on polymer X on days 4 and 8 were immunoblotted with antibodies specific to FN1, integrin ɑ5, JAK1, JAK2, GP130, p-STAT3, total STAT3, and α-tubulin. (**E**) Image obtained by culturing STAT3 GFP-negative cell line transfected with either non-target siRNA or si*FN1*, cultured on polymer X for 8 days using IncuCyte system. Scale bar, 200 μm. (**F**) Cell lysates from SKOV3 cancer cells transfected with either non-target siRNA or si*JAK2*, cultured on polymer X on days 4 and 8, were immunoblotted with antibodies specific to JAK1, JAK2, GP130, p-STAT3, total STAT3, and α-tubulin. (**G**) Cell lysates from SKOV3 cancer cells transfected with either non-target siRNA or si*ITGA5*, cultured on TCP or polymer X on days 4 and 8, were immunoblotted with antibodies specific to integrin ɑ5, JAK1, JAK2, GP130, p-STAT3, total STAT3, and α-tubulin.

**Figure 3 biomedicines-10-02684-f003:**
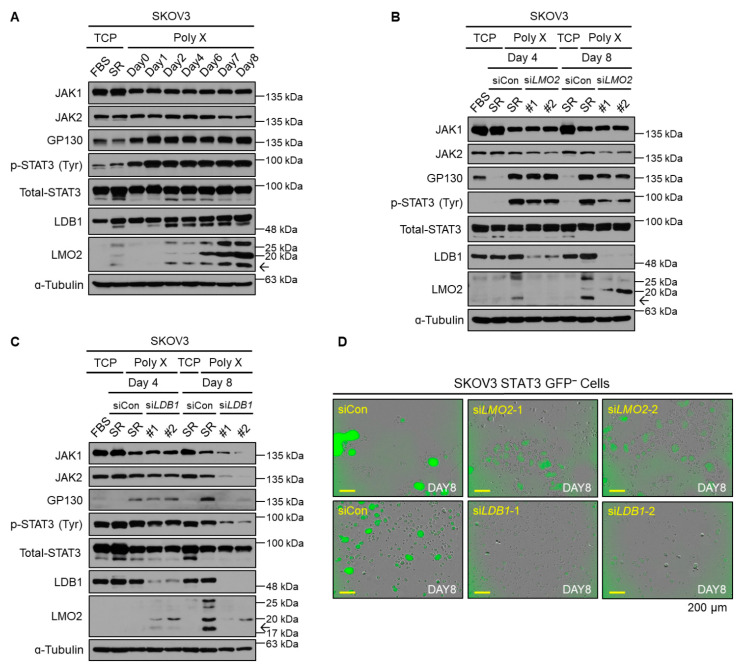
Long-term activation of STAT3 signaling via LMO2-LDB1 complex. (**A**) Cell lysates from SKOV3 cancer cells cultured on polymer X by date were immunoblotted with antibodies specific to JAK1, JAK2, GP130, p-STAT3, total STAT3, LMO2, LDB1, and α-tubulin. (**B**) Cell lysates from SKOV3 cancer cells transfected with either non-target siRNA or si*LMO2*, cultured on TCP or polymer X on days 4 and 8, were immunoblotted with antibodies specific to JAK1, JAK2, GP130, p-STAT3, total STAT3, LMO2, LDB1, and α-tubulin. (**C**) Cell lysates from SKOV3 cancer cells transfected with either non-target siRNA or si*LDB1*, cultured on TCP or polymer X on days 4 and 8, were immunoblotted with antibodies specific to JAK1, JAK2, GP130, p-STAT3, total STAT3, LMO2, LDB1, and α-tubulin. (**D**) Image obtained by culturing STAT3 GFP-negative cell line transfected with either non-target siRNA or si*LMO2* andsi*LDB1*, cultured on polymer X for 8 days using IncuCyte system. Scale bar, 200 μm. The arrow means LMO2.

**Figure 4 biomedicines-10-02684-f004:**
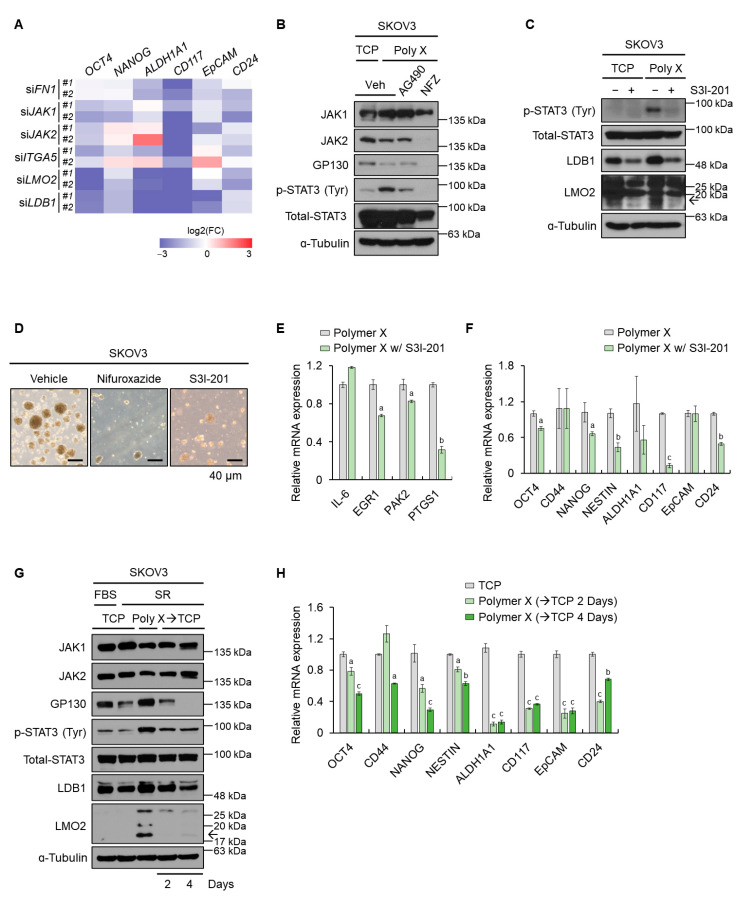
Polymer X induced-tumor spheroids acquired cancer stem-like properties via STAT3 signaling. (**A**) A heatmap showing the mRNA levels of stemness markers obtained by knocking down each gene as displayed and performing real-time PCR. (**B**) Cell lysates from SKOV3 cancer cells treated with the STAT3 inhibitors, AG490 and nifuroxazide (NFZ), cultured on TCP or polymer X were immunoblotted with antibodies specific to JAK1, JAK2, GP130, p-STAT3, total STAT3, and α-tubulin. (**C**) Cell lysates from SKOV3 cancer cells treated with the STAT3 inhibitor, S3I-201, cultured on TCP or polymer X were immunoblotted with antibodies specific to p-STAT3, total STAT3, LDB1, LMO2, and ɑ-tubulin. (**D**) Images of tumor spheroids cultured on polymer X while administering STAT3 inhibitors. Scale bar, 40 μm. (**E**) The mRNA levels of STAT3 target genes in SKOV3 cancer cells treated with S3I-201 or vehicle were determined by real-time PCR. Data are expressed as mean ± SEM. Two-tailed Student’s *t*-test was used to analyze the statistical significance between each group (*n* = 3 for each group). a: *p* < 0.05; b: *p* < 0.01.(**F**) The mRNA levels of each indicated gene in SKOV3 cancer cells treated with S3I-201 or vehicle were determined by real-time PCR. Data are expressed as mean ± SEM. Two-tailed Student’s *t*-test was used to analyze the statistical significance between each group (*n* = 3 for each group). a: *p* < 0.05; b: *p* < 0.01; and c: *p* < 0.001. (**G**) Cell lysates from SKOV3 cancer cells cultured on TCP and polymer X, and those cultured on polymer X and transferred to the TCP on days 2 and 4 were immunoblotted with antibodies specific to JAK1, JAK2, GP130, p-STAT3, total STAT3, LMO2, LDB1, and α-tubulin. (**H**) mRNA levels of each indicated gene in SKOV3 cancer cells cultured on TCP and polymer X, and those cultured on polymer X and transferred to the TCP on days 2 and 4 were determined by real-time PCR. Data are expressed as mean ± SEM. Two-tailed Student’s *t*-test was used to analyze the statistical significance between each group (*n* = 3 for each group). a: *p* < 0.05; b: *p* < 0.01; and c: *p* < 0.001. The arrow means LMO2.

**Figure 5 biomedicines-10-02684-f005:**
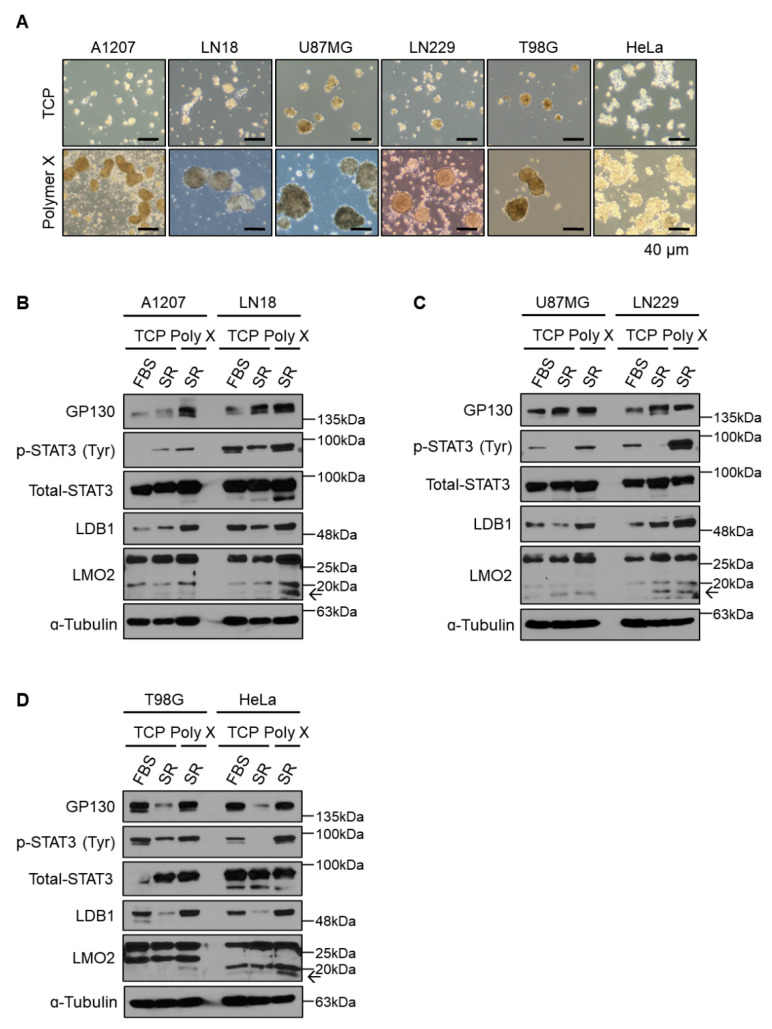
General application of polymer X using various cancer cell lines. (**A**) Images of tumor spheroids cultured in each indicated cell line on TCP or polymer X for 8 days. Scale bar, 40 μm. (**B**) Cell lysates from A1207 and LN18 GBM cells cultured on TCP or polymer X for 8 days were immunoblotted with antibodies specific to GP130, p-STAT3, total STAT3, LMO2, LDB1, and α-tubulin. (**C**) Cell lysates from U87MG and LN229 GBM cells cultured on TCP or polymer X for 8 days were immunoblotted with antibodies specific to GP130, p-STAT3, total STAT3, LMO2, LDB1, and α-tubulin. (**D**) Cell lysates from T98G GBM cells and HeLa cells cultured on TCP or polymer X for 8 days were immunoblotted with antibodies specific to GP130, p-STAT3, total STAT3, LMO2, LDB1, and ɑ-tubulin. The arrow means LMO2.

## Data Availability

RNA sequencing data are available at the National Center for Biotechnology Information Gene Expression Omnibus (GEO) data repository with the accession code (GSE213872). Any other relevant data are available from the lead contact upon request.

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
