# Peer review of "Polymer Thin Film Promotes Tumor Spheroid Formation via JAK2-STAT3 Signaling Primed by Fibronectin-Integrin α5 and Sustained by LMO2-LDB1 Complex"

_biomedicines, 2022, doi:10.3390/biomedicines10112684_

Round 1

Reviewer 1 Report

I have gone through the manuscript. The topic is indeed interesting but the description of JAK1 and JAK2 as described in western blots is confusing. Information provided by Authors is scattered and in piecemeals. Authors described the information insufficiently in earlier western blots of JAK1 and JAK2. Therefore, keeping in view the fact that wholistically moves around JAK/STAT signaling. Therefore authors should be crystal clear in the description related to JAK1 and JAK2 and specifically how JAK2 levels were increased. Once fixed, the information will be interconnected with increased JAK2 and p-STAT3. Authors should scrupulously explain the figure 2 and 3 in the paragraphs and also fully elucidate how different treatments regulated the activation of JAK1 and JAK2 in figure 2 and 3. 

Reviewer 2 Report

The study demonstrates the effect of polymer X in CSC characteristics. The 3.3. Long-term activation of STAT3 signaling via LMO2-LDB11 complex may be revised to confirm the description of the expression levels of JAK2, since it does not seem to be decreased at day2 in Figure 3A.

Author Response

Responses to the comments of the reviewer #2

Comment: The study demonstrates the effect of polymer X in CSC characteristics. The 3.3. Long-term activation of STAT3 signaling via LMO2-LDB11 complex may be revised to confirm the description of the expression levels of JAK2, since it does not seem to be decreased at day2 in Figure 3A.

Response: We sincerely appreciate the reviewer’s helpful comment. As the reviewer pointed out, the protein level of JAK2 in Figure 2C decreased from day 1, but the protein level of JAK2 in Figure 3A decreased after 4 days. As a result of continuous detection of JAK2 during the experiment, the expression pattern itself is always the same, which increases in the early phase (day 1~4) and then decreases over time. However, it seems to vary slightly depending on the number of cells and culture context at that time. Therefore, the expression of decreasing with the exact date, such as 'at day 2' can be confusing enough. We modified that the expression levels of JAK2 was increased from day 1, but decreases over time. Please see page 10, lines 5-8 of the revised manuscript (below).

Page 10, lines 5-8 in Results

3.3. Long-term activation of STAT3 signaling via LMO2-LDB1 complex

In our study, polymer X-cultured cells displayed fibronectin expression was in-creased from day 1 with phosphorylation of STAT3, whereas the expression levels of JAK2 was increased from day 1, but gradually decreased over time.

Round 2

Reviewer 1 Report

Looks in good form now. 

Author Response

Responses to the comments of the reviewer #1

Comment: Moderate English changes required.

Response: We sincerely appreciate the reviewer's helpful comments. All minor typos and misrepresentations, including those pointed out by the academic editor, have been corrected throughout the manuscript.
